# Relationship between diabetic macular edema and choroidal layer thickness

**Hiroaki Endo[1], Satoru Kase[2]\*, Mitsuo Takahashi[1], Michiyuki Saito[1], Masahiko Yokoi[3], Chisato Sugawara[1], Satoshi Katsuta[1], Susumu Ishida[2], Manabu Kase[1]**

**1** Department of Ophthalmology, Teine Keijinkai Hospital, Sapporo, Japan, **2** Department of Ophthalmology, Faculty of Medicine and Graduate School of Medicine, Hokkaido University, Sapporo, Japan, **3** Teine Yokoi Eye Clinic, Sapporo, Japan

\* kaseron@med.hokudai.ac.jp

**Data Availability Statement:** All relevant data are within the manuscript and Supporting Information files.

## Abstract

### Purpose

To investigate the relationship between diabetic macular edema (DME) and the choroidal layer thickness in diabetic patients.

### Methods

This is a retrospective observation study. Three hundred eighteen eyes of 159 diabetes mellitus (DM) patients and age-matched 100 eyes of 79 healthy controls were enrolled. DME was defined as over 300 μm in the central retinal subfield of the Early Treatment Diabetic Retinopathy Study (ETDRS) grid sector. The central choroidal thickness (CCT), as well as inner and outer layers were determined based on enhanced depth imaging (EDI)–OCT. Diabetic patients with/without systemic diabetic treatments (DT) at the start of this study was defined as DT+ and DT−, respectively. The number of eyes examined was 62 and 256 eyes in DME+and DME−groups, respectively. DM patients were further subdivided into 4 groups with/without DME and DT; DME+DT+(35 eyes), DME−DT+(159 eyes), DME+DT−(27 eyes), and DME−DT−group (97 eyes). Multiple comparisons on CCT layers including control and each DM group were statistically examined.

### Results

The total CCT layer was 254±83, 283±88, and 251±70 μm in the control, DME+, and DME−group, respectively. A total CCT layer in DME+was significantly thicker than the DME−group ($P < 0.05$). The outer CCT layer was 195±75, 222±83, and 193±63 μm in the control, DME+, and DME−group, respectively. The outer CCT layer in DME+ was significantly thicker than the DME−group ($P < 0.05$). In the subdivided groups, the total CCT layers in the control, DME+DT+, DME−DT+, DME+DT−and DME−DT−groups were 254±83, 274±88, 247±66, 290±84 and 258±75 μm, respectively. The outer CCT layers in each group were 195±75, 214±83, 189±58, 228±77, and 201±70 μm, respectively. Total CCT and the outer layer in DME+DT−was significantly thicker than the DME−DT+group (each $P < 0.05$). In contrast, there was no significant difference in inner layer between the groups.

**Funding:** The author(s) received no specific funding for this work.

**Competing interests:** The authors have declared that no competing interests exist.

## Conclusions

The total and outer CCT layers of diabetic eyes were significantly thickened in the DME +DT–as compared with the DME–DT+group, suggesting that CCT may be related to the pathology of DME.

## Introduction

Diabetic macular edema (DME) is a major cause of visual impairment in the working age population [1]. The pathogenesis of DME is mainly caused by retinal vascular hyperpermeability. The choroid is an important vascular tissue responsible for blood supply to the retinal pigment epithelium and photoreceptor cells, which plays pivotal roles in the metabolic exchange to the avascular fovea [2]. It is indisputable that the choroid takes part in not only the physiological homeostasis in the outer retina, but also the pathology of many retinal diseases including diabetic retinopathy (DR). Experimental and clinical analyses have proved that choroidal vascular disorders in diabetes might involve the pathogenesis of DR [3,4]. Histopathological changes in choroidal vessels in diabetic patients are similar to retinal vascular changes seen in DR such as increased vascular tortuosity, extravascular hemorrhage, microaneurysms, nonperfusion areas, vasodilatation and stenosis as well as the neovascularization [3]. It has also been reported that protein liquid leakage to the stromal dropout and choroid interstitium possibly reflects ischemia and vascular hyperpermeability, which is observed in the retinal vasculature of DR as well [5].

With recent advances in imaging technologies, the choroids have been mainly evaluated by indocyanine green angiography [4] and laser doppler flowmetry [6,7]. In more recent years, the visualization of the choroid was improved by the enhanced depth imaging-optical coherence tomography (EDI-OCT), which enables ophthalmologists to obtain macular cross-sectional images with high quality and resolution [8]. Clinical evaluation in the choroids of DM eyes has been conducted; however, the results examined regarding choroid thickness were various, such as thickened, thinned, or no changes [9]. Similarly, the previous results of the choroidal thickness measurement in DME eyes demonstrated thinned [10–13], thickened [14,15], and no change [16–18]. We have investigated patients with DR by dividing them into systemic DM treatment and DR severity, and concluded that the central choroid thickness (CCT) was significantly reduced in early DR without treatment [19]. More recently, we showed that the changes in CCT of DR patients might result from outer choroidal layer thickness [20]. Thus, the most important pathological change in the choroid in DR may extend not only to the inner layer including choriocapillaris but also to the outer layer of the choroid [9]. However, little is known about the relationship between the choroidal layer thickness of DME eyes and DM treatment.

The aim of this study is to investigate the relationship between DME with/without diabetes treatment and the thickness of choroidal layer in diabetic patients.

## Methods

### Subjects and clinical study protocol

The participants provided their verbal informed consent to participate in this study. In fact, the institutional review committee of Teine Keijinkai hospital (IRB) approved this consent procedure and also gave an opportunity to refuse the participation anytime to the participants

by disclosing the study protocol on thewebsite: http://www.keijinkai.com/teine/about/efforts/feature/". This retrospective observation study was approved by the IRB, and informed consent was obtained from all subjects (patients diagnosed with type 2 diabetes and healthy controls). This study followed the principles of the Declaration of Helsinki. Three hundred and eighteen eyes with 159 DM patients comprising 107 males (67.3%) and 52 females (32.7%) (mean age 61.3±11.8 years), and 100 eyes in 79 normal subjects without DM (31 male (39.2%) and 48 females (60.8%); mean age 59.8±13.9 years) were enrolled in this study. All subjects underwent tests including a best-corrected visual acuity (BCVA, decimal scale), Goldmann applanation tonometry, biomicroscopy of the anterior segment using a slit lamp examination, and ophthalmoscopy of the posterior segment, and axial length measured using optical biometry (IOL Master Zeiss; Jena, Germany) and EDI-OCT (Cirrus HD–OCT, Carl Zeiss Meditec) during December 2013 to April 2018. Exclusion criteria were eyes with a history of other ocular disease, previous topical treatments including retinal laser photocoagulation, local vascular endothelial growth factor (VEGF) and/or triamcinolone acetonide injection, or any ocular surgeries including vitrectomy, cataract or glaucoma surgery. Moreover, eyes with spherical power greater than –5 diopters, cylindrical astigmatism with cylinder power greater than 3 diopters, and ocular axial length being greater than 26 mm were excluded. Age-matched healthy subjects were recruited from consecutive populations scheduled for routine ophthalmic examinations to check refractive errors. The selection criteria for control subjects were the same as those for DM subjects, except for the absence of diabetes.

First, all the DM patients were divided into diabetics with/without DME (DME+/–) and compared those with the control group. DM patients further were divided into two groups based on information regarding systemic DM treatment (DT) situation: patients (DM-treated group: DT+) who received systemic pharmacotherapy with oral hypoglycemic or subcutaneous insulin therapy for DM managed by the attending physician, or who did not receive systemic treatments for DM (DM-untreated group: DT–) until the start of this study. Finally, DM patients were classified into 4 groups according to with/without DME and DT+/–; DME+DT+, DME–DT+, DME+DT–, and DME–DT–group. Those 4 groups were eventually compared with the control group. In addition, DME + group was classified into three groups based on the disease stage of DR, that is, international severity classification; mild / moderate non-proliferative DR (mNPDR), severe NPDR (sNPDR), and PDR. Furthermore, DME was classified into 3 groups according to the configuration of DME based on the previous report by Otani et al. [21]; sponge-like retinal swelling (SLRS), cystoid macular edema (CME), and serous retinal detachment (SRD). Each classified DR group was compared with control group, respectively.

## Retinal and choroidal thickness measurements using EDI-OCT imaging

OCT facilitates objective quantification of retinal thickness. Central retinal thickness (CRT) was obtained by automatic retina thickness map in the Early Treatment Diabetic Retinopathy Study (ETDRS) grid sector. DME was defined as over 300 μm in the central retinal subfield of the ETDRS grid sector. As described in the previous report, this study analyzed choroid tomograms obtained by the EDI-OCT method spectral domain OCT (Cirrus HD OCT; Carl Zeiss Meditec, Inc., Dublin, CA, USA) [19]. The EDI-OCT scan included 9 mm horizontal lines centered on the fovea, obtained by experienced technicians. They have confirmed that signal strength of all the measured values was more than 6 out of 10. Therefore, images with a signal strength of less than 6 were excluded from this study. Anatomically, the choroidal vascular layers are made up of three layers: choriocapillaris, medium vascular layer (Sattler's layer) and large vessel layer (Haller's layer); however, the boundary of the three-layer structure is not

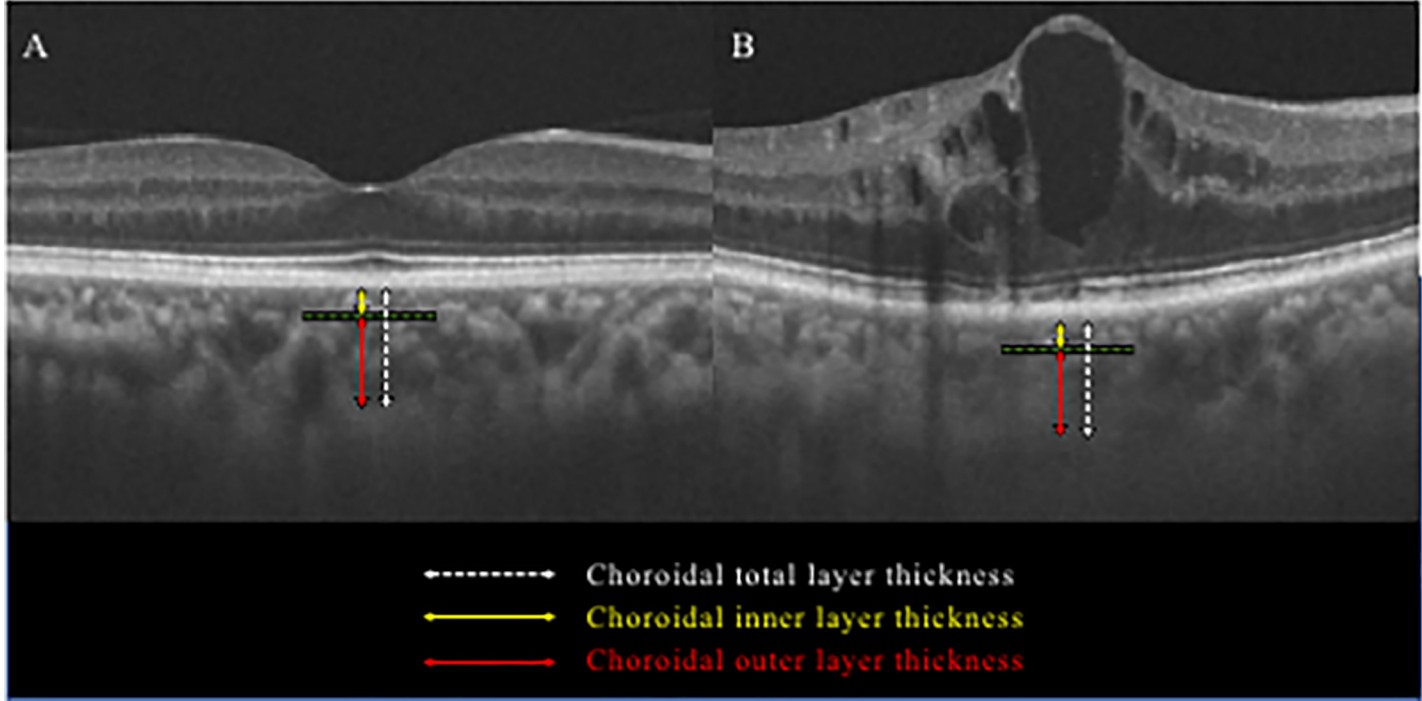

**Fig 1.** Identification of choroid layer thicknesses by EDI-OCT in a healthy control (A) and diabetic macular edema (B). The choroid outer layer thickness (red line) was measured from the inner boundary of the choroid scleral junction to the innermost point (green line) of the choroidal large blood vessel observed closest to the fovea centralis. The choroid inner layer thickness (yellow line) was obtained by subtracting the choroidal outer layer thickness (red line) from the total choroid layer thickness (white line).

clear compared to the retina. For this reason, CCT data from the total choroid layer, inner layer (choriocapillaris + medium vascular layer), and outer layer was manually identified based on findings of EDI-OCT horizontal scan through the fovea, as reported previously (Fig 1) [22]. To identify the measurement position of the choroid thickness, the position where the foveal bulge was confirmed, or the position where the vertical height of the DME was the highest, was defined as the central retinal thickness, and then the choroid thickness was measured beneath the positions. Two evaluators (H.E. and C.S.) independently assessed OCT images in a masked fashion regarding the subject's clinical information.

## Statistical analyses

Age, best-corrected visual acuity, refractive error, intraocular pressure, ocular axial length, hemoglobin A1c (HbA1c), systolic blood pressure (SBP), diastolic blood pressure (DBP), estimated glomerular filtration rate (eGFR), total cholesterol (TC), high density lipoprotein cholesterol (HDL-C), low density lipoprotein cholesterol (LDL-C), triglyceride (TG), between DM patient and control groups were analyzed using Steel test. Retinal thickness, choroid layer thickness between DM patients and control groups were calculated using Kruskal-Wallis test and analyzed using Steel-Dwass test. Data of choroid layer thickness obtained by examiners were analyzed for relative reliability and absolute reliability. Relative and absolute reliability were examined using Spearman's rank correlation coefficient and Bland-Altman plot, respectively. In order to ensure measurement accuracy, the absolute agreement proved by the Bland-Altman plot between measurements is applied for fixed and proportional biases. Correlation between visual acuity in the DME eye and retinal and choroid layer thicknesses was examined

**Table 1. Patient's characteristics of control and diabetes mellitus groups.**

|  | Control group | DM group | |
|---|---|---|---|
|  |  | DME+ | DME− |
| Total number of eyes | 100 | 62 | 256 |
| Age (years) | 59.8±13.9 | 59.3±12.0 | 61.7±11.7 |
| BCVA–logMAR (snellen) | −0.02±0.16 (20/19) | 0.29±0.34 (20/39) ** | 0.06±0.22 (20/23) ** |
| Spherical error (diopter) | −0.13±1.56 | −0.40±1.59 | −0.39±1.66 |
| IOP (mmHg) | 14.6±3.1 | 15.3±3.9 | 15.2±3.1 |
| Axial length (mm) | 23.80±0.83 | 23.57±0.88 | 23.65±0.93 |
| DM duration (years) | − | 9.7±7.3 | 10.9±9.0 |
| HbA1c (%) | 5.6±0.4 | 8.33±2.17** | 8.61±2.19** |
| SBP (mmHg) | 122±16 | 139±22** | 137±21** |
| DBP (mmHg) | 75±13 | 77±11 | 78±12 |
| eGFR (mℓ/min/1.73 m$^2$) | 76±16 | 69±32 | 75±29 |
| TC (mg/dl) | 208±34 | 195±44 | 193±49* |
| HDL-C (mg/dl) | 63±14 | 53±16 | 49±14** |
| LDL-C (mg/dl) | 109±25 | 119±36 | 112±37 |
| TG (mg/dl) | 97±46 | 140±58* | 172±124** |

* :$P<0.05$

** :$P<0.01$ vs. control group

DM, diabetes mellitus; DME, diabetic macular edema; BCVA, best corrected visual acuity; IOP, intraocular pressure; HbA1c, hemoglobin A1c; SBP, systolic blood pressure; DBP, diastolic blood pressure; eGFR, estimated glomerular filtration rate; diastolic blood pressure; TC, total cholesterol; HDL-C, high density lipoprotein cholesterol; LDL-C, low density lipoprotein cholesterol; TG, triglyceride

using Spearman's rank correlation coefficient. Data were presented as mean±standard deviation. In all studies, a P value of less than 0.05 was considered statistically significant, which was determined using the statistical software commercially available (SPSS version 21.0).

## Results

The clinical characteristics of the study group are summarized in Table 1. Three hundred and eighteen eyes with 159 DM patients comprising 107 males (67.3%) and 52 females (32.7%) (mean age 61.3±11.8 years), and 100 eyes in 79 normal subjects without DM (31 male (39.2%) and 48 females (60.8%); mean age 59.8±13.9 years) were enrolled in this study. The number of DME + or DME− eyes was 62 eyes and 256 eyes, respectively. There were no significant differences in age, equivalent sphere, intraocular pressure, axial length, diabetes duration, DBP, eGFR, and LDL-C between DM group and control group. HbA1c, SBP and TG were significantly higher in DM group than in control group, respectively (each, P<0.01). On the other hand, TC and HDL-C were significantly lower in DM group than in control group (P<0.05).

The number of DT−patients with systemic DM treatment discontinuation or without history of DM treatments was 27 and 35 cases, respectively. The number of eyes in the DME + DT +, DME-DT +, DME + DT-, and DME-DT- groups classified into 4 groups according to with/without DME and DT were 35, 159, 27 and 97, respectively (Table 2). There was no significant difference in age, equivalent spherical power, intraocular pressure, ocular axial length, duration of diabetes, SBP, DBP, eGFR, TC, HDL-C, LDL-C, and TG in each DM group. On the other hand, the HbA1c value was significantly higher in the DT−group (10.0±2.4%) than in the DT+group (7.5±1.3%) (P < 0.001 in Table 1). There were no significant differences in

**Table 2. Patient's characteristics of control and DM-treated/untreated groups.**

| | Control group | DM-treated group | | DM-untreated group | |
|---|---|---|---|---|---|
| | | DME+ | DME | DME+ | DME |
| Total number of eyes | 100 | 35 | 159 | 27 | 97 |
| Age (years) | 59.8±13.9 | 61.8±10.8 | 63.8±10.9 | 56.1±12.6 | 58.4±12.1 |
| BCVA–logMAR (snellen) | −0.02±0.16 (20/19) | 0.31±0.34 (20/41)** | 0.05±0.21 (20/22)** | 0.27±0.34 (20/37)** | 0.10±0.25 (20/25)** |
| Spherical error (diopter) | −0.13±1.56 | −0.73±1.21 | −0.68±1.56 | −1.01±1.72 | −0.45±1.32 |
| IOP (mmHg) | 14.6±3.1 | 14.6±3.8 | 15.3±3.1 | 14.3±3.7 | 15.0±3.1 |
| Axial length (mm) | 23.80±0.83 | 23.50±1.01 | 23.61±0.94 | 23.67±1.01 | 23.70±0.97 |
| DM duration (years) | – | 9.9±8.0 | 11.6±9.3 | 9.1±6.0 | 9.2±7.7 |
| HbA1c (%) | 5.6±0.4 | 7.2±1.0** | 7.6±1.3** | 9.8±2.4** | 10.2±2.4** |
| SBP (mmHg) | 122±16 | 140±20* | 134±18* | 138±24 | 140±23** |
| DBP (mmHg) | 75±13 | 81±10 | 76±12 | 73±10 | 80±12 |
| eGFR (mℓ/min/1.73 m$^2$) | 76±16 | 73±23 | 70±24 | 67±37 | 80±32 |
| TC (mg/dl) | 208±34 | 191±35 | 188±47** | 198±50 | 200±52 |
| HDL-C (mg/dl) | 63±14 | 56±12 | 48±12** | 49±20 | 50±16** |
| LDL-C (mg/dl) | 109±25 | 112±35 | 104±32 | 130±34 | 122±40 |
| TG (mg/dl) | 97±46 | 141±57 | 168±122* | 139±59 | 176±128** |

\* :$P<0.05$

\*\*:$P<0.01$ vs. control group

DM, diabetes mellitus; DME, diabetic macular edema; BCVA, best corrected visual acuity; IOP, intraocular pressure; HbA1c, hemoglobin A1c; SBP, systolic blood pressure; DBP, diastolic blood pressure; eGFR, estimated glomerular filtration rate; diastolic blood pressure; TC, total cholesterol; HDL-C, high density lipoprotein cholesterol; LDL-C, low density lipoprotein cholesterol; TG, triglyceride

age, equivalent sphere, intraocular pressure, axial length, diabetes duration, DBP, eGFR, and LDL-C between DM group and control group. SBP and TG were significantly higher in DM group than in control group, respectively (each, $P<0.01$). On the other hand, TC and HDL-C were significantly lower in DM group than in control group ($P<0.05$).

The reproducibility of the CCT measurements for total and outer layers were examined in normal and diabetic eyes (S1 Table). In all the layers tested, the intraclass correlation coefficient (ICC) exceeded 0.9, which reflected high reproducibility (S1 Table). Next, with the limit of agreement determined by Bland- Altman plot for choroidal thickness in normal and diabetic eyes, the systematic bias in the measured values was visualized with dots (S1 Fig). In order to judge the presence of fixed bias, a 95% confidence interval of the average of the difference between two measurements was obtained. The 95% confidence interval was −0.56 to 2.20, −1.22 to 1.74, −0.44 to 1.27, and −0.86 to 1.09, respectively, proving no significant fixed bias in choroidal layers in normal and DM eyes (S1 Table). Next, in order to judge the presence of proportional bias, the significant correlation in the difference and the average measured between the corresponding two groups was tested. The correlation was R = 0.007 ($P = 0.94$), R = 0.099 ($P = 0.33$), R = −0.109 ($P = 0.053$), and R = −0.101 ($P = 0.07$), respectively, indicating no significant proportional bias in choroidal layers in normal and DM eyes (S1 Table). These data overall suggest that there was no particular tendency to cause the difference between the evaluators.

In this study, the number of eyes examined was 62 and 256 eyes in DME+ and DME– groups, respectively. The total CCT layer was 254±83, 283±88, and 251±70 μm in the control, DME+, and DME–group, respectively (Fig 2B). A total CCT layer in DME+was significantly thicker than the DME–group ($P < 0.05$). Next, the inner CCT layer was 59±13, 60±12, and 57 ±13 μm in the control group, DME+, and DME–groups, respectively (Fig 2C). There was no

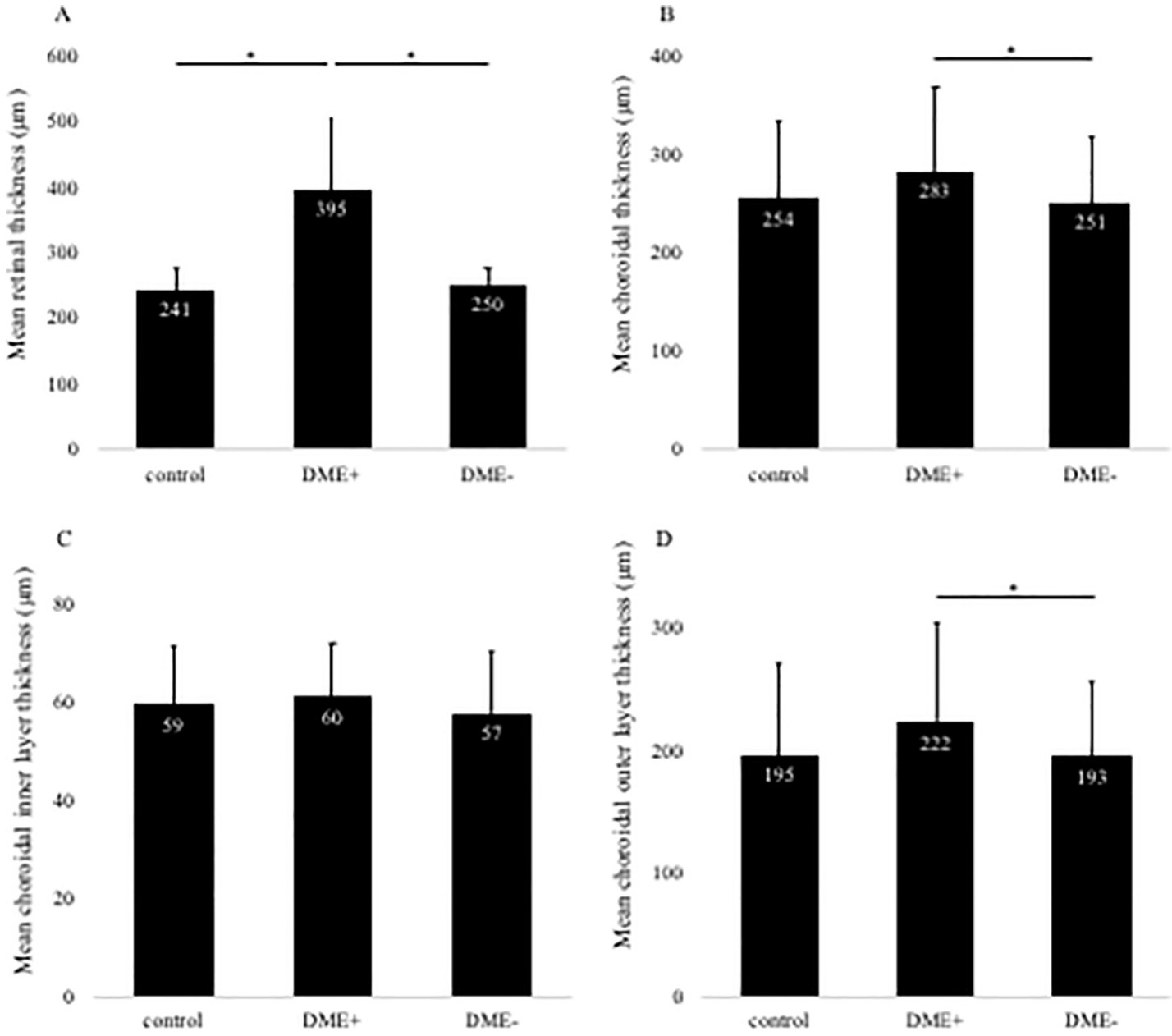

**Fig 2.** Changes in retinal thickness (A) and total choroidal layer thickness (B) with or without diabetic macular edema (DME). DME + group shows significantly thicker retinal thickness than control and DME−groups (A). DME + group reveals a significantly thicker choroidal thickness than DME- group (B). Changes in choroidal inner layer thickness (C) and outer layer thickness (D) with or without DME. There was no significant difference in all groups about inner layer (C). In contrast, DME+group reveals a significantly thicker choroidal thickness than DME- group (D). Mean retinal and choroidal thickness are shown in the three groups. Error bars represent standard deviation. The asterisk (*) indicates a significant difference (P < 0.05).

significant difference among all groups in the inner CCT layer. The outer CCT layer was 195 ±75, 222±83, and 193±63 μm in the control, DME+, and DME−group, respectively (Fig 2D). The outer CCT layer in DME+ was significantly thicker than the DME−group ($P < 0.05$).

In this study, the number of eyes examined was 161 and 124 eyes in the DT + and DT-groups, respectively. The retinal thickness was significantly thicker in the diabetic group than in the control group (Fig 3A). Next, there was no significant difference in total CCT, inner

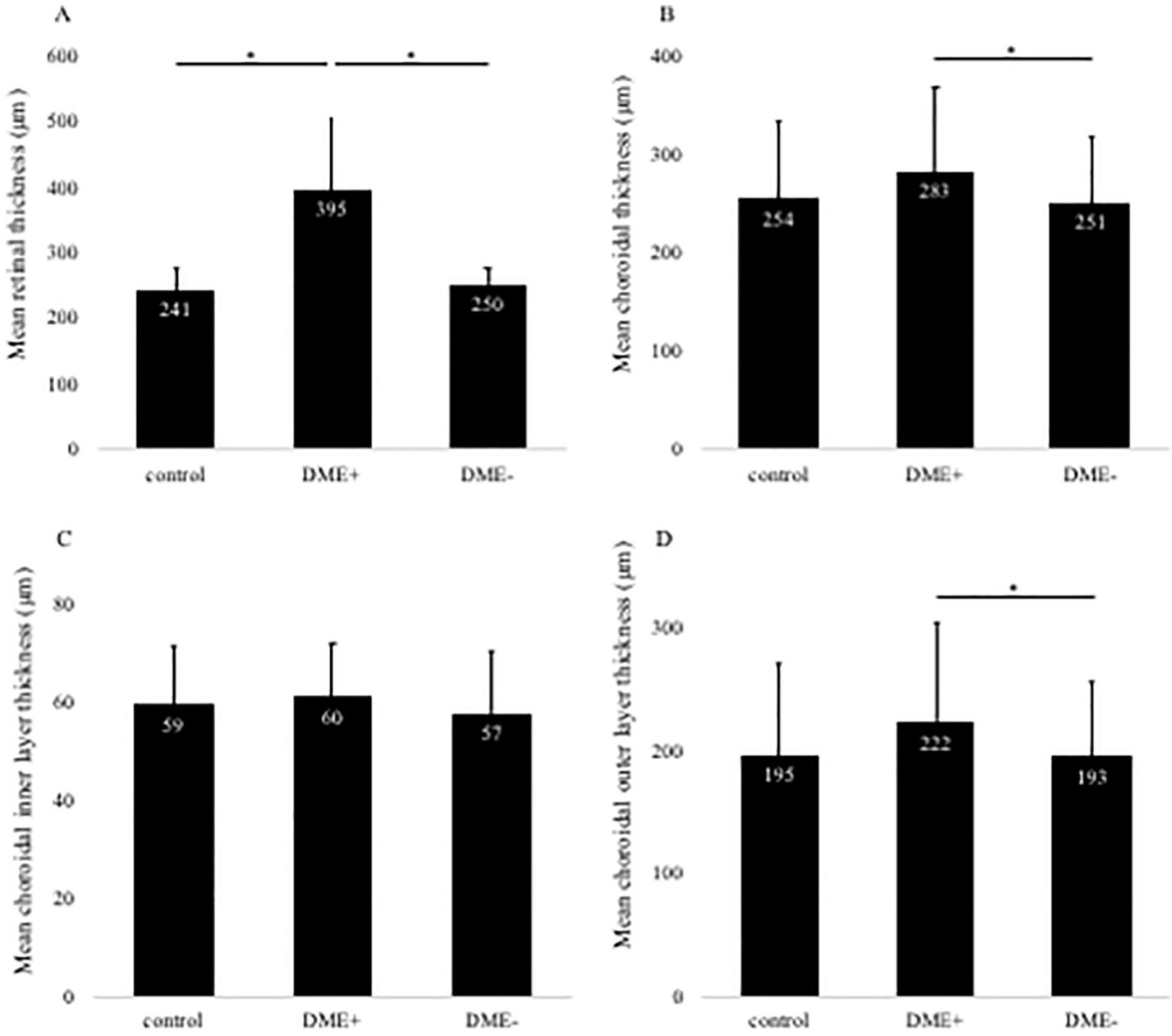

**Fig 3.** Changes in retinal thickness (A) and total choroidal layer thickness (B) with or without systemic diabetes treatment (DT). DT+and DT−group reveals a significantly thicker retinal thickness than control group (A). There was no significant difference in all groups about choroidal layer (B). Changes in choroidal inner layer thickness (C) and outer layer thickness (D) with or without DT. There was no significant difference in all groups about inner (C) and outer layer (D). Mean retinal and choroidal thickness are shown in the three groups. Error bars represent standard deviation. The asterisk (*) indicates a significant difference (P < 0.05). The double asterisk (**) indicates a significant difference (P < 0.01).

CCT, and outer CCT between the control and the diabetic group (Fig 3B–3D). The characteristics of the control group and DM 4 groups are shown in Table 2. The number of eyes in DT +group was 35 and 159 in DME+and DME−, respectively, and the number of eyes in DT−was 27 and 97 in DME+and DME−, respectively. Total CCT layers were 254±83, 274±88, 247±66, 290±84, and 258±75 μm for control, DME+and DME−in DT+, and DME+ and DME−in DT−, respectively (Fig 4B). The total CCT layer in DME+DT−was significantly thicker than the

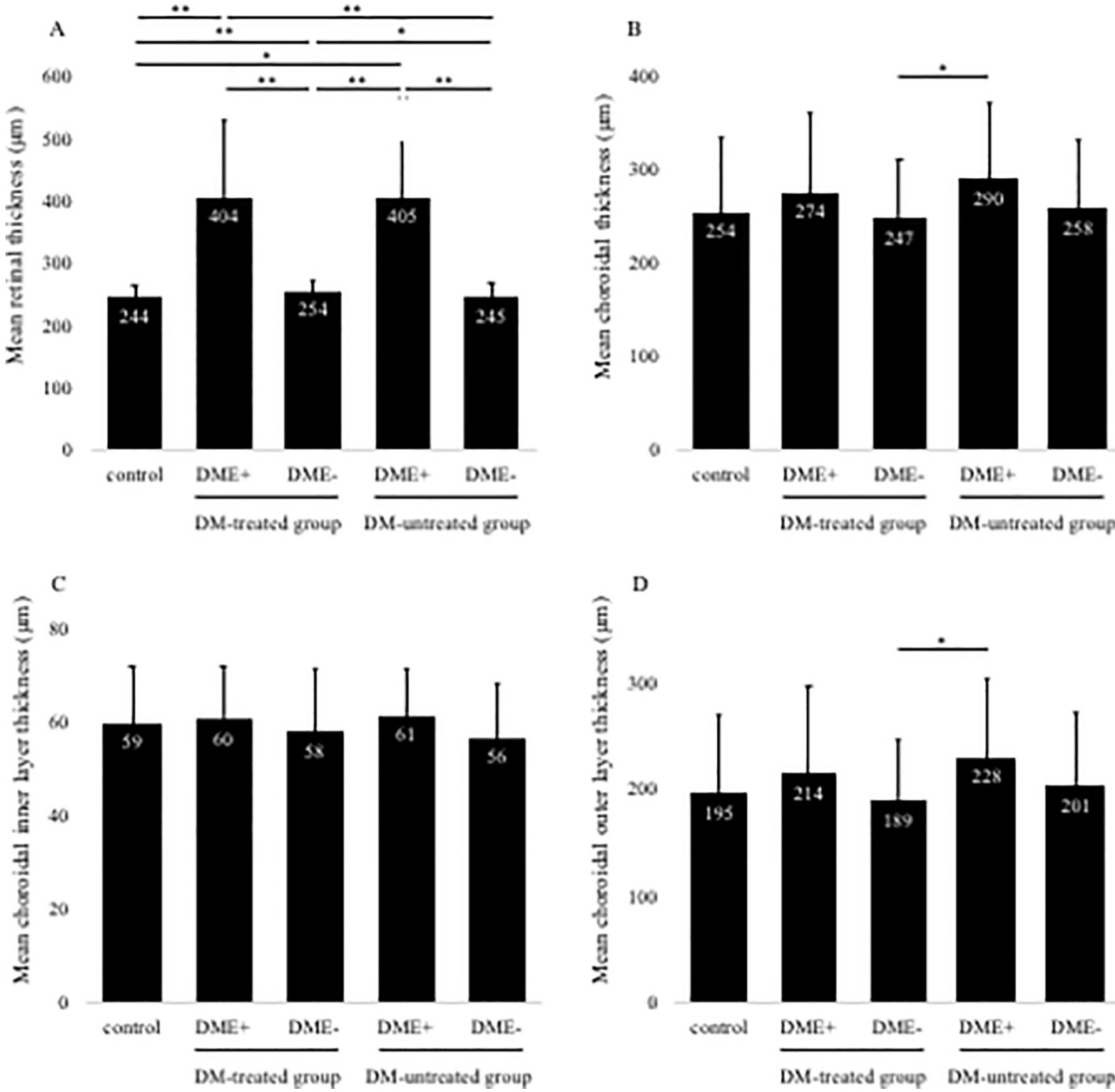

**Fig 4.** Changes in retinal thickness (A) and choroidal total layer thickness (B) in diabetic eyes based on the history of systemic treatments. Retinal thickness was significantly thicker in DME+groups regardless of DM treatments, and DME- groups than the control group (A). In contrast, DM-untreated DME + group shows a significantly thicker choroidal thickness than DM-treated DME- group (B). Changes in choroidal inner layer thickness (C) and outer layer thickness (D) in diabetic eyes. There was no significant difference in all groups in inner layer (C). DM-untreated DME + group reveals a significantly thicker outer choroidal thickness than DM-treated DME- group (D). Mean retinal and choroidal thicknesses are shown in the five groups. Error bars represent standard deviation. The asterisk (*) indicates a significant difference (P < 0.05). The double asterisk (**) indicates a significant difference (P < 0.01).

DME–DT+group (P < 0.05). Next, inner CCT layer was 59±13, 60±12, 58±14 and 61±12 and 56±12 µm for control, DME+and DME–in DT+, and DME+ and DME–in DT–groups,

**Table 3. Choroid layer thickness of Control and DR stage groups.**

| | Control group | DR stage groups | | |
| --- | --- | --- | --- | --- |
| | | mNPDR | sNPDR | PDR |
| Total number of eyes | 100 | 19 | 17 | 26 |
| Retinal thickness in the central subfield | 244±23 | 355±133** | 381±136** | 389±207** |
| Choroid total layer thickness | 254±83 | 266±122 | 307±59 | 279±81 |
| Choroidal inner layer thickness | 59±13 | 56±10 | 67±10* | 59±11 |
| Choroidal outer layer thickness | 195±75 | 210±106 | 239±56 | 220±76 |

*:$P<0.05$

**:$P<0.01$ vs. control group

DR, diabetic retinopathy; DME, diabetic macular edema; mNPDR, mild/moderate non-proliferative DR; sNPDR, severe NPDR

respectively (Fig 4C). There was no significant difference between all groups in the inner CCT layer. Finally, outer CCT layer was 195±75, 214±83, 189±58 and 228±77, and 201±70 μm for control, DME+and DME–in DT+, and DME+ and DME–in DT–, respectively (Fig 4D). The outer CCT layer in DME+DT–was significantly thicker than the DME–DT+ group ($P < 0.05$), whereas there was no significant difference between DT + and DT–groups in outer CCT layer of DME+eyes.

Choroidal layer thickness by each stage of DR in DME + group and by each type of DME is shown in Tables 3 and 4, respectively. In stage of DR, inner CCT layer was significantly thicker in sNPDR compared to the control group (Table 3, $P < 0.05$). In each type of DME, choroidal layer thickness was not significantly different as compared with the control group. CRT was significantly thicker in DME+groups regardless of DM treatments, and DME- groups than the control group (Fig 4A). Correlation between BCVA and parameters on the OCT image in the DME+ group had a significant positive correlation with CRT ($R = 0.50$, $P < 0.01$), whereas there was no correlation with each choroid layer thickness (Table 5).

## Discussion

In this study, the total CCT layer was significantly thicker in the DME+ than in the DME– group in all DME cases examined ($P < 0.05$). Previous studies demonstrated that the choroidal thickness of untreated DME eyes was thinned [10–13], thickened [14,15], or no change [16–18], and the consequence was different among the researchers. One of reasons in such different results on CCT might be different inclusion criteria even in untreated DME eyes

**Table 4. Choroid layer thickness of Control and DME subtype groups.**

| | Control group | DME subtype groups | | |
| --- | --- | --- | --- | --- |
| | | SLRS | CME | SRD |
| Total number of eyes | 100 | 24 | 27 | 11 |
| Retinal thickness in the central subfield | 244±23 | 327±26** | 422±88** | 528±261** |
| Choroid total layer thickness | 254±83 | 294±109 | 268±70 | 294±70 |
| Choroidal inner layer thickness | 59±13 | 60±12 | 61±12 | 60±9 |
| Choroidal outer layer thickness | 195±75 | 234±102 | 207±65 | 234±66 |

*:$P<0.05$

**:$P<0.01$ vs. control group

DR, diabetic retinopathy; DME, diabetic macular edema; SLRS, sponge-like retinal swelling; CME, cystoid macular edema; SRD, serous retinal detachment

**Table 5. The relationship between best corrected visual acuity, retinal thickness, and choroidal thickness in the diabetic macular edema eyes.**

| Variables (n = 62) | 1. BCVA–logMAR | 2. Retinal thickness in the central subfield | 3. Choroidal total layer thickness | 4. Choroidal inner layer thickness | 5. Choroidal outer layer thickness |
|---|---|---|---|---|---|
| 1. BCVA–logMAR | – | 0.50** | -0.24 | -0.12 | -0.25 |
| 2. Retinal thickness in the central subfield | 0.50** | – | -0.05 | 0.002 | -0.04 |
| 3. Choroidal total layer thickness | -0.24 | -0.05 | – | 0.58** | 0.99** |
| 4. Choroidal inner layer thickness | -0.12 | 0.002 | 0.58** | – | 0.51** |
| 5. Choroidal outer layer thickness | -0.25 | -0.04 | 0.99** | 0.51** | – |

**:$P < 0.01$

BCVA, best corrected visual acuity

examined. For example, studies showing thinning of CCT in DME eyes were lack of eyes with PDR in the study subjects, and the number of eyes examined was small [10–13]. On the other hand, studies showing no change in CCT included eyes with all the DR stages including PDR. In addition, CCT is influenced by the patient background of the study subject, systemic or topical treatment history, differences in treatment methods, and differences between races. Indeed, CCT has been suggested to be involved with local factors of the eye in DR. Several authors have shown that CCT could be affected by panretinal photocoagulation [23], intravitreal anti-VEGF therapy [24], and intravitreal triamcinolone acetonide injection [25]. As a matter of fact, these have been shown to be effective treatments for patients with DME. In this study, we collected cases without any ocular treatment history, and consequently showed significant thickening of the total CCT layer in the DME eye compared to the eye without DME. Our results are consistent with previous publications including the report by Kim et al., looking at DME eyes without local treatments [14,15].

Mechanisms underlying the thickened choroid in DME eyes has yet to be elucidated. VEGF is an important cytokine that mediates vascular hyper-permeability, and an increase in the level of VEGF protein concentration has also been observed in the DME eyes [26]. In addition, the choroidal thickness was significantly reduced after anti-VEGF therapy in DME eyes [27,28]. Okamoto et al. examined the choroidal blood flow before and after intravitreal ranibizumab injection into DME eyes using laser speckle flowgraphy, and showed that the choroidal blood flow significantly decreased one month after the treatment [15]. An animal study has demonstrated that the choroidal vascular development and homeostasis are highly dependent on VEGF [29]. Taken together, these results suggested that alteration of choroidal thickness and the blood flow in DME depends on VEGF. Therefore, the association between DME and CCT seen in this study may have correlated with increased VEGF expression, leading to increased choroidal blood flow and subsequent CCT thickening. Although it is still unknown whether the onset of DME affects the cause or the result of CCT thickening, further studies are needed to clarify the mechanisms.

Recently, it has been reported that many systemic and physiological conditions involving hemodynamics influence DME and choroidal thickness [30]. Large-scale epidemiological studies indicated that elevated HbA1c increased the risk of the onset of DME [31,32]. Hwang et al. evaluated macular thickness before and after hemodialysis in diabetic patients with end-stage renal disease, and found a significant decrease of choroidal thickness in DME eyes after hemodialysis [33]. In the FIELD study that examined the effect of fenofibrate for the treatment

of dyslipidemia in type 2 diabetic patients, the risk of DME progression decreased by 31% in the fenofibrate group compared to the placebo group [34]. These results suggest a complex mechanistic association between DME and systemic factors, providing evidence to manage DME patients by systemic controls. We previously examined subdivided DR patients with/ without systemic DM treatment or DR severity, and demonstrated that CCT was significantly reduced in mild/moderate nonproliferative DR without DM treatment [19]. In addition, we showed that changes in CCT of DM eyes possibly depended on choroidal outer layer thickness [20]. These indicated that chronic hyperglycemia may be a further deteriorating factor for choroidal microcirculation disorders.

A previous study analyzing the choroid layer thickness reported that the thickness of the inner layer combining the choriocapillaris and medium vascular layers significantly decreased in the PDR and DME eyes [11]. On the other hand, the most significant changes observed in the choroid not only limited the choriocapillaris, but also extended to larger vessels [9]. In this study, the outer choroidal layer thickness in all the DM patients examined was significantly greater in the DME+ than in the DME–group. In the previous report, VEGF protein concentrations increased in DME eyes [35], which suggest the reason why the outer CCT layer increased in the DME+group. Previously we demonstrated that there was no significant correlation between CCT and DR stages including outer layer thickness in the DM treatment group, and that HbA1c value was significantly lower in treatment group than in DM untreated group [19,20]. These results suggest that systemic DM therapy might play an important role in the stabilization of the outer layer thickness in the choroid. HbA1c levels in this study was also significantly lower in DT+group than in DT–group (P < 0.01, Table 1). In addition, the outer choroidal thickness of DM patients in this study changed depending on with/without DT and DME, and was significantly thinner in the DME–DT+group than in the DME+DT–group. However, there was no significant difference between DT+group and DT–group in total and outer CCT layers of DME+patients, indicating systemic DM treatment alone is not an independent factor involving the choroidal thickness.

Therefore, we speculated the mechanisms underlying significantly thickened outer layer in DME+DT–group as follows: 1) chronic hyperglycemic condition caused by untreated DM deteriorated choroidal microcirculatory system in DME eyes, 2) more increase in VEGF protein associated with DME, led to vasodilatation of outer choroid layer, increased choroidal blood flow and/or vascular hyperpermeability, 3) increased accumulation of several diabetes-related proteins including advanced glycation end products in choroidal tissues, as shown in DM donor eyes with immunohistochemistry [36]. However, the mechanisms underlying thickened inner choroidal layers in sNPDR-DME+ group compared to the controls are unknown. In fact, the major histopathological findings in the inner layer of DM eyes were occlusion of choriocapillaris [3]. Therefore, in order to validate the clinicopathological correlation, it is necessary to investigate morphological changes in choroidal vessels and/or stromal components using recent imaging techniques. Future studies including the lumen/stroma morphology with binarization methods may clarify the pathology of the inner and outer choroidal layers in DME.

It is well known that DME can cause refractory vision disorder in DM patients. In our study, there was a significant negative correlation between BCVA in the DME and CRT determined by OCT findings, but BCVA was not correlated with choroid layer thickness (Table 5). Yiu et al. evaluated the relevance to changes in visual and anatomical outcomes during 6 months following anti-VEGF treatment for DME eyes, and reported that there was no association between BCVA and CCT [27]. In contrast, Eliwa et al. showed a significantly negative correlation between BCVA and CCT in the DME eyes [13]. The differences between the previous and the current studies may be related to different DR stages examined. They limited the

NPDR in the DME group [13], whereas we included the DME patients having from NPDR to PDR. In addition, CCT in DME patients in their study showed a significant thinning compared to healthy controls, whereas our results showed rather thickening. These results suggest that CCT in DME might correlate with patients' vision in the early stage of DR.

The current study has several limitations. First, we measured the layer thickness including the outer thickness manually. It is indisputable to ensure data reproducibility between evaluators. Since Wong et al. have reported that the presence of thickened choroid and subretinal fluid may affect the reproducibility [37], these factors may involve measurements of the thickness of the choroidal layer in this study. Second, the thickness of the choroid layer was evaluated only in the central choroid in this study. It has been shown that parafoveal choroidal thickness may be different in each case [38]. Recently, a binarization method can be used to assess choroidal vascular changes in diabetic eyes [39]; therefore, further study is needed to clarify a wide range of layer thicknesses. Third, classification of systemic diabetes treatment was performed with or without medication treatment. However, detailed information on systemic treatments for diabetes mellitus could not be obtained. Therefore, differences in drug medications given in each subject might affect choroidal thickness. Fourth, changes in choroidal thickness may also be associated with choroidal blood flow. However, since data about indocyanine green angiography and optical coherence tomography angiography are not available in this study, further studies are needed to clarify the association between choroidal thickness and the circulation. Fifth, this study was able to adjust the local confounding factor that could affect the choroid structure. However, systolic blood pressure and serum lipids levels were significantly higher in the DM group than normal controls. These systemic confounding factors might affect the acquisition of data regarding choroidal structure. Finally, the layer thickness was examined based on EDI-OCT imaging in this study. Further studies will also prove differences of data obtained by EDI-OCT or swept source OCT in diabetic eyes.

In conclusion, the total and outer CCT layer of the DM eye was significantly thickened in the DME+DT−group compared to the DME−DT+group. These results indicated that CCT may be related to the pathology of DME.

## Supporting information

**S1 Fig. Band-Altman blot analysis at total choroidal thickness and outer thickness in normal and diabetic eyes.** Total choroidal thickness (A) and outer choroidal thickness (B) in normal eyes. Total choroidal thickness (C) and outer choroidal thickness (D) in diabetic eyes. Solid line indicates the average mean difference, while dotted line shows 95% confidence limit of agreement. There is no specific trend to result in the difference between raters.
(TIFF)

**S1 File. The support file is available at the file "20191106 Data in supporting information. xlsx.**
(XLSX)

**S1 Table. Inter-examiner reliability of choroidal segmentation for control and diabetes mellitus group.**
(DOCX)

## Author Contributions

**Conceptualization:** Satoru Kase, Mitsuo Takahashi, Susumu Ishida.

**Data curation:** Hiroaki Endo, Chisato Sugawara.

**Formal analysis:** Satoru Kase, Mitsuo Takahashi, Chisato Sugawara.

**Investigation:** Satoru Kase.

**Methodology:** Hiroaki Endo, Satoru Kase, Michiyuki Saito.

**Project administration:** Hiroaki Endo, Masahiko Yokoi.

**Software:** Hiroaki Endo.

**Supervision:** Mitsuo Takahashi, Chisato Sugawara, Satoshi Katsuta, Susumu Ishida, Manabu Kase.

**Validation:** Manabu Kase.

**Visualization:** Michiyuki Saito.

**Writing – original draft:** Satoru Kase.

**Writing – review & editing:** Mitsuo Takahashi, Michiyuki Saito, Masahiko Yokoi.

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
