## [Decision Letter · Decision Letter 0]

28 Oct 2019

PONE-D-19-26407

Relationship between diabetic macular edema and choroidal layer thickness

PLOS ONE

Dear Dr Kase,

Thank you for submitting your manuscript to PLOS ONE. After careful consideration, we feel that it has merit but does not fully meet PLOS ONE’s publication criteria as it currently stands. Therefore, we invite you to submit a revised version of the manuscript that addresses the points raised during the review process.

Both reviewers found this work interesting but in need of improvement. Especially about how concussions are drawn form associations that cannot determine causality.  I believe the points raised can be addressed and we would look forward to a substantially revised version 

We would appreciate receiving your revised manuscript by Dec 12 2019 11:59PM. To enhance the reproducibility of your results, we recommend that if applicable you deposit your laboratory protocols in protocols.io, where a protocol can be assigned its own identifier (DOI) such that it can be cited independently in the future. For instructions see: http://journals.plos.org/plosone/s/submission-guidelines#loc-laboratory-protocols

We look forward to receiving your revised manuscript.

Kind regards,

Demetrios G. Vavvas

Academic Editor

PLOS ONE

Journal Requirements:

a) Did participants provide their written or verbal informed consent to participate in this study?

3. Thank you for stating the following financial disclosure: No

Please provide an amended Funding Statement that declares *all* the funding or sources of support received during this specific study (whether external or internal to your organization) as detailed online in our guide for authors at http://journals.plos.org/plosone/s/submit-now.  Please state what role the funders took in the study.  If any authors received a salary from any of your funders, please state which authors and which funder. If the funders had no role, please state: "The funders had no role in study design, data collection and analysis, decision to publish, or preparation of the manuscript."

No

Reviewers' comments:

Reviewer's Responses to Questions

**Comments to the Author**

1. Is the manuscript technically sound, and do the data support the conclusions?

Reviewer #1: Partly

Reviewer #2: Partly

2. Has the statistical analysis been performed appropriately and rigorously? 

Reviewer #1: I Don't Know

Reviewer #2: Yes

3. Have the authors made all data underlying the findings in their manuscript fully available?

Reviewer #1: Yes

Reviewer #2: Yes

4. Is the manuscript presented in an intelligible fashion and written in standard English?

Reviewer #1: No

Reviewer #2: Yes

5. Review Comments to the Author

Reviewer #1: The authors investigated the relationship between diabetic macular edema (DME) and choroidal layer thickness. They also revealed the significance of the systemic condition in diabetes mellitus (DM) patients in CT evaluation. It could be an important contribution. The data is well collected and presented clearly. However, overall, this paper seems somewhat disorganized, especially in Methods and Results section. I have the following concerns.

First and foremost, please recheck and follow our submission guidelines such as double-spaced layout, line number included, etc. (See https://journals.plos.org/plosone/s/submission-guidelines). And I recommend

Major concerns,

(1) Page 4: Subjects and Clinical study protocol: The number of subjects, sex, and eyes should be moved to Result section.

(2) Page 4: Exclusion criteria: Did this study include the subjects with another ocular disease and surgery (cataract, glaucoma, etc.)?

(3) Page 4: Please describe normal subjects clearly. How did the authors collect and choose normal subjects? Age-matched? Were they all healthy subjects? No ocular history?

(4) In Results section, these sections should be revised to improve clarity. I suggest the authors describe the data and analyses of three main groups (control, DME+, DME-) with a new Table (separated from Table 1) and Figure 3 in the 1st paragraph. After that, the subgroup data and analyses should be shown. This flow is the same order the authors stated in Methods section.

(5) Page 6, 1st Paragraph: Why did the authors do the statistical analyses for groups with AND without control (Why twice)? Is it a proper statistical approach?

(6) Table 1: The footnote showed that asterisk means statistically significance "versus control." If so, the authors should state using Steel test (not Steel-Dwass test) for this analysis in Methods section.

(7) Page 7, 2nd Paragraph, and Table 2: This reproducibility analysis is not needed in this paper. The authors did not discuss it at all (they mentioned the reproducibility as the study limitation, though). And above all, it is not essential data for this study aim. I recommend that it should be deleted or moved to Supplementary data.

(8) Page 8, 4th Paragraph: This paragraph is confusing. Please rewrite the paragraph for our readers to easily understand. And how were these subgroups' backgrounds? Was there no significant difference among the subgroups (including control) in their backgrounds?

(9) Table 3: The information should be presented in two separate tables, one focusing on the stage of diabetic retinopathy and the other on type of DME).

Minor concerns,

(10) Was the flare measured in this study? The flare data may support the possible significance of VEGF in DME and choroidal layer thickness. (The relationship between the flare value and the concentration of VEGF in aqueous was reported before)

Reviewer #2: The authors aim to investigate the relationship between diabetic macular edema (DME) and the choroidal layer thickness in diabetic patients. The findings are interesting; however, some points need to be clarified.

1) Although the authors found -in each DM group- that there was no significant difference in systolic blood pressure, this latter was significantly higher in DM group than in control group (P<0.01). Do they think that this could have influenced their results? Please make a comment on it.

2) Authors indicated in the methods that data from the total choroid layer, inner layer (choriocapillaris + medium vascular layer), and outer layer was manually identified based on findings of EDI-OCT horizontal scan through the fovea, as reported previously [22]. Since patients with DME usually do not have foveal depression, which scan did the authors include to measure choroidal layers?

3) I know they considered excluded images with signal strength less than 6. Nevertheless, I believe that in this kind of study (manual measurements) it is important to statistically correct all the OCT images for signal strength index. Also, what is about the presence of significant motion artifacts?

4) I would suggest the authors to include just one eye per patient, as they correlated several parameters referred to a subject (for example: duration of diabetes, age, blood pressure and so on) to retinal parameters that are measured separately in a single eye. Although the numbers will be more limited, it would be more correct to use a single eye in the statistical analyses.

5) The found that in stage of DR, inner CCT layer was significantly thicker in severe non-proliferative diabetic retinopathy compared to the control group (P < 0.05). How many patients with DME in this group?

6) Did they include patients with Type 2 DM? Type 2 and type 1 DM? These latter usually are younger and have fewer confounding factors or systemic comorbidities.

7) Based on their finding, the authors stated that systemic DM treatment alone is not an independent factor involving the choroidal thickness. Also, they speculated that one of the mechanisms underlying significantly thickened outer layer in DME＋DT-group might be correlated to chronic hyperglycemic condition caused by untreated DM.

One could argue why patients with DME- DT- (97 patients and second largest group) do not present any choroidal statistically significant difference as an early sign of the deteriorated choroidal microcirculatory system.

Please make a comment on it.

Minor:

Page 5, please delete the double references #19

6. PLOS authors have the option to publish the peer review history of their article (what does this mean?). If published, this will include your full peer review and any attached files.

Reviewer #1: No

Reviewer #2: No

---

## [Author Response · Author response to Decision Letter 0]

7 Nov 2019

For your convenience, the reviewers’ comments will appear by a single underline, and then our reply will follow the comments.

Reviewers' comments #1 to the Author:

Reviewer #1: 

Major concerns 

(1) Page 4: Subjects and Clinical study protocol: The number of subjects, sex, and eyes should be moved to Result section.

Thank you so much for your comments.

We moved the number of subjects, gender, and eyes listed in the Methods section to the appropriate location in the Results section.

(2) Page 4: Exclusion criteria: Did this study include the subjects with another ocular disease and surgery (cataract, glaucoma, etc.)?

Thank you so much for your comments.

The exclusion criteria are listed in the Method section as follows.

Page 6, 1st Paragraph: Exclusion criteria were eyes with a history of other ocular disease, previous topical treatments including retinal laser photocoagulation, local vascular endothelial growth factor (VEGF) and/or triamcinolone acetonide injection, or any ocular surgeries including vitrectomy, cataract or glaucoma surgery. Moreover, eyes with spherical power greater than －5 diopters, cylindrical astigmatism with cylinder power greater than 3 diopters, ocular axial length being greater than 26 mm, and other fundus diseases were excluded.

Therefore, subjects with other eye diseases and surgical history are not included. 

(3) Page 4: Please describe normal subjects clearly. How did the authors collect and choose normal subjects? Age-matched? Were they all healthy subjects? No ocular history?

Thank you so much for your comments.

In order to clarify the selection criteria for healthy controls, the following sentence was added in the revised version.

Page 6, 1st Paragraph: Age-matched healthy subjects were recruited from consecutive patients scheduled for routine ophthalmic examinations to check refractive errors. The selection criteria for control subjects were the same as those for DM subjects, except for the absence of diabetes. The statements were added in the Methods section in the revised version.

(4) In Results section, these sections should be revised to improve clarity. I suggest the authors describe the data and analyses of three main groups (control, DME+, DME-) with a new Table (separated from Table 1) and Figure 3 in the 1st paragraph. After that, the subgroup data and analyses should be shown. This flow is the same order the authors stated in Methods section.

Thank you so much for your comments.

In order to improve the clarity of the results, the data and analysis of the three main groups (control, DME +, DME-) were prepared as a new table 1. Therefore, the data and analysis corresponding to the results section were added. 

Page 8, 1st paragraph: The clinical characteristics of the study group are summarized in Table 1. Three hundred and eighteen eyes with 159 DM patients comprising 107 males (67.3%) and 52 females (32.7%) (mean age 61.3±11.8 years), and 100 eyes in 79 normal subjects without DM (31 male (39.2%) and 48 females (60.8%); mean age 59.8±13.9 years) were enrolled in this study. The number of DME + or DME− eyes was 62 eyes and 256 eyes, respectively. There were no significant differences in age, equivalent sphere, intraocular pressure, axial length, diabetes duration, DBP, eGFR, and LDL-C between DM group and control group. HbA1c, SBP and TG were significantly higher in DM group than in control group, respectively (each, P<0.01). On the other hand, TC and HDL-C were significantly lower in DM group than in control group (P<0.05).

(5) Page 6, 1st Paragraph: Why did the authors do the statistical analyses for groups with AND without control (Why twice)? Is it a proper statistical approach?

Thank you so much for your comments.

First, the study group was simplified and limited to variables that could affect the results. Therefore, the analysis was performed in three groups: control, DME +, and DME-. CCT is known to be affected by systemic and/or local factors. In this study, we collected and examined cases that had no history of ophthalmic treatments. In addition, we examined diabetic eyes including DME which merged with all the stages of DR. Indeed, previous publications reporting the thinning of CCT included DME eyes without PDR, and the sample size was small. On the other hand, the other studies showed thickening and no change of CCT, which included all stages including PDR. Furthermore, we investigated the relationship between DME and choroidal thickness in diabetic patients in a new classification with and without diabetes mellitus treatments to investigate the impact of systemic diabetes treatment. In this way, we think that it is a proper statistical approach.

(6) Table 1: The footnote showed that asterisk means statistically significance "versus control." If so, the authors should state using Steel test (not Steel-Dwass test) for this analysis in Methods section.

Thank you so much for your comments. As you suggested, the following factors were tested by Steel test or Steel-Dwass test, appropriately.

The revised version also added the following statements in the method section.

Page 8: Age, best-corrected visual acuity, refractive error, intraocular pressure, ocular axial length, hemoglobin A1c (HbA1c), systolic blood pressure (SBP), diastolic blood pressure (DBP), estimated glomerular filtration rate (eGFR), total cholesterol (TC), high density lipoprotein cholesterol (HDL-C), low density lipoprotein cholesterol (LDL-C), triglyceride (TG), between DM patient and control groups were analyzed using Steel test. Retinal thickness, choroid layer thickness between DM patient and control groups were calculated using Kruskal-Wallis test and analyzed using Steel-Dwass test.

(7) Page 7, 2nd Paragraph, and Table 2: This reproducibility analysis is not needed in this paper. The authors did not discuss it at all (they mentioned the reproducibility as the study limitation, though). And above all, it is not essential data for this study aim. I recommend that it should be deleted or moved to Supplementary data.

Thank you so much for your comments.

As indicated, we did not discuss reproducibility analysis. However, we believe that reproducibility in measuring choroid thickness is important to clarify the reliability of data.

Therefore, we moved the initial Table 3 and Figure 2 to Supplementary table 1 and Supplementary figure 1, respectively.

(8) Page 8, 4th Paragraph: This paragraph is confusing. Please rewrite the paragraph for our readers to easily understand. And how were these subgroups' backgrounds? Was there no significant difference among the subgroups (including control) in their backgrounds?

Thank you so much for your comments.

It is well known that there is a systemic confounding factors in measuring choroidal thicknesses. Therefore, high systolic blood pressure and serum lipids in the DM group may have influenced the results.

Therefore, the revised version added the following in the discussion section.

Page 19, 2th Paragraph: Fifth, this study was able to adjust the local confounding factor that could affect the choroid structure. However, systolic blood pressure and serum lipids were significantly higher in the DM group than normal controls. These systemic confounding factors might affect the acquisition of data regarding choroidal structure.

(9) Table 3: The information should be presented in two separate tables, one focusing on the stage of diabetic retinopathy and the other on type of DME).

Thank you so much for your comments.

A separated table was created, focusing on the stage of the DR and the type of DME (Tables 3 ang 4).

Minor concerns,

(10) Was the flare measured in this study? The flare data may support the possible significance of VEGF in DME and choroidal layer thickness. (The relationship between the flare value and the concentration of VEGF in aqueous was reported before)

Thank you so much for your comments.

It is known that the breakdown of the blood-retina barrier in diabetic patients is correlated with a dysfunction of the blood-aqueous barrier. In fact, several researches showed that flare levels change after photocoagulation and anti-VEGF therapy in patients with DME. However, although interesting, there are no reports showing significant correlation between flare levels and choroidal thickness in DME.

In this study, the flare level of DME patients was not investigated, and its relationship could not be examined.

This will be an issue for a further study.

Reviewer #2:

1) Although the authors found -in each DM group- that there was no significant difference in systolic blood pressure, this latter was significantly higher in DM group than in control group (P<0.01). Do they think that this could have influenced their results? Please make a comment on it.

Thank you so much for your comments.

It has been pointed out that there is a systemic confounding factors in the choroidal thickness. Therefore, high systolic blood pressure and serum lipids in the DM group may have influenced the results. Therefore, in relation to reply to the reviewer 1’s comment, the revised version added the following in the discussion section.

Page 19, 2th Paragraph: Fifth, this study was able to adjust the local confounding factor that could affect the choroid structure. However, systolic blood pressure and serum lipids were significantly higher in the DM group than normal controls. These systemic confounding factors can affect the acquisition of data regarding choroidal structure.

2) Authors indicated in the methods that data from the total choroid layer, inner layer (choriocapillaris + medium vascular layer), and outer layer was manually identified based on findings of EDI-OCT horizontal scan through the fovea, as reported previously [22]. Since patients with DME usually do not have foveal depression, which scan did the authors include to measure choroidal layers?

Thank you so much for your comments.

The position where the foveal bulge was confirmed, or the position where the vertical height of the DME was the highest were defined as the central retinal thickness, and then the choroid thickness was measured beneath the positions.

Therefore, the following was added to the method section.

Page 7: To identify the measurement position of the choroid thickness, the position where the foveal bulge was confirmed, or the position where the vertical height of the DME was the highest were defined as the central retinal thickness, and then the choroid thickness was measured beneath the positions. 

3) I know they considered excluded images with signal strength less than 6. Nevertheless, I believe that in this kind of study (manual measurements) it is important to statistically correct all the OCT images for signal strength index. Also, what is about the presence of significant motion artifacts?

Thank you so much for your comments.

It is a well-known fact that low-quality OCT images affect analysis. In this study, obtained images with a signal strength of 6 or more were included. Motion artifacts were resolved by using a fundus tracking system. Therefore, the following was added to the method section.

Page 7: Therefore, images with a signal strength of less than 6 were excluded from this study. In addition, motion artifacts were removed by the fundus tracking system.

4) I would suggest the authors to include just one eye per patient, as they correlated several parameters referred to a subject (for example: duration of diabetes, age, blood pressure and so on) to retinal parameters that are measured separately in a single eye. Although the numbers will be more limited, it would be more correct to use a single eye in the statistical analyses.

Thank you so much for your comments.

In the previous reports, there are many reports analyzing choroidal thicknesses using both eyes on the topic of DME (1. Unsal E et al. Choroidal thickness in patients with diabetic retinopathy. Clin Ophthalmol. 2014. 2. Kim JT et al. Changes in choroidal thickness in relation to the severity of retinopathy and macular edema in type 2 diabetic patients. Invest Ophthalmol Vis Sci. 2013. 3. Rewbury R et al. Subfoveal choroidal thickness in patients with diabetic retinopathy and diabetic macular oedema. Eye (Lond). 2016.).

In this study, the proportion of binocular DME cases with different forms of edema in each eye was 20%. This fact indicates that the DME even in the same case may have different pathology and different severity in each eye. In addition, this is a retrospective study, and if we selected and examined the one eye in binocular cases, bias may occur in DME showing different conditions in both eyes. Thus, prospective studies may be more appropriate for unilateral studies. Therefore, we think that the current study design including both eyes is meaningful.

5) The found that in stage of DR, inner CCT layer was significantly thicker in severe non-proliferative diabetic retinopathy compared to the control group (P < 0.05). How many patients with DME in this group?

Thank you so much for your comments.

The sNPDR-DME group included 17 eyes from 14 patients.　

Therefore, the following was added to the discussion section.

Page 18: However, the mechanisms underlying thickened inner choroidal layers in sNPDR-DME+ group compared to the controls are unknown. In fact, the major histopathological findings in the inner layer of DM eyes were occlusion of choriocapillaris [3]. Therefore, in order to validate the clinicopathological correlation, it is necessary to investigate morphological changes in choroidal vessels and/or stromal components using recent imaging techniques.

6) Did they include patients with Type 2 DM? Type 2 and type 1 DM? These latter usually are younger and have fewer confounding factors or systemic comorbidities.

Thank you so much for your comments.

All diabetic patients examined in this study were type 2 DM. Hence, subjects of type 1 DM patients may be examined for future investigations. The revised version added the following statements in the method section.

Page 6, 1st paragraph: This retrospective observation study was approved by the institutional review committee of Teine Keijinkai hospital, and informed consent was obtained from all subjects (patients diagnosed with type 2 diabetes and healthy controls).

7) Based on their finding, the authors stated that systemic DM treatment alone is not an independent factor involving the choroidal thickness. Also, they speculated that one of the mechanisms underlying significantly thickened outer layer in DME＋DT-group might be correlated to chronic hyperglycemic condition caused by untreated DM.

One could argue why patients with DME- DT- (97 patients and second largest group) do not present any choroidal statistically significant difference as an early sign of the deteriorated choroidal microcirculatory system.

Please make a comment on it.

Thank you so much for your comments.

The choroid thickness may be regulated by VEGF produced in the eye. In fact, it has been reported that the intraocular VEGF protein concentration is higher than the serum concentration in DME eyes (Funatsu, H., et al., Increased levels of vascular endothelial growth factor and interleukin-6 in the aqueous humor of diabetics with macular edema. Am J Ophthalmol, 2002). Therefore, from the standpoint that both ocular local and systemic factors are related to CCT, it is considered that the single factor DT alone did not show a significant change, as we showed no significant changes between DME+DT- and DME+DT-. 

Minor:

Page 5, please delete the double references #19

Thank you so much for your comments.

The corresponding part was corrected.

Thank you

---

## [Decision Letter · Decision Letter 1]

4 Dec 2019

Relationship between diabetic macular edema and choroidal layer thickness

PONE-D-19-26407R1

Dear Dr. Kase,

We are pleased to inform you that your manuscript has been judged scientifically suitable for publication and will be formally accepted for publication once it complies with all outstanding technical requirements.

With kind regards,

Demetrios G. Vavvas

Academic Editor

PLOS ONE

Additional Editor Comments (optional):

Reviewers' comments:

Reviewer's Responses to Questions

**Comments to the Author**

1. If the authors have adequately addressed your comments raised in a previous round of review and you feel that this manuscript is now acceptable for publication, you may indicate that here to bypass the “Comments to the Author” section, enter your conflict of interest statement in the “Confidential to Editor” section, and submit your "Accept" recommendation.

Reviewer #1: All comments have been addressed

Reviewer #2: All comments have been addressed

2. Is the manuscript technically sound, and do the data support the conclusions?

Reviewer #1: Yes

Reviewer #2: Yes

3. Has the statistical analysis been performed appropriately and rigorously? 

Reviewer #1: I Don't Know

Reviewer #2: Yes

4. Have the authors made all data underlying the findings in their manuscript fully available?

Reviewer #1: Yes

Reviewer #2: No

5. Is the manuscript presented in an intelligible fashion and written in standard English?

Reviewer #1: Yes

Reviewer #2: No

6. Review Comments to the Author

Reviewer #1: (No Response)

Reviewer #2: Afeter the revision, authors have addressed all my comments and the manuscript quality has been improved.

7. PLOS authors have the option to publish the peer review history of their article (what does this mean?). If published, this will include your full peer review and any attached files.

Reviewer #1: No

Reviewer #2: No

---

## [Editor Report · Acceptance letter]

10 Dec 2019

PONE-D-19-26407R1 

Relationship between diabetic macular edema and choroidal layer thickness 

Dear Dr. Kase:

I am pleased to inform you that your manuscript has been deemed suitable for publication in PLOS ONE. Congratulations! Your manuscript is now with our production department. 

With kind regards,

on behalf of

Dr. Demetrios G. Vavvas 

Academic Editor

PLOS ONE